# Feasibility of a Home-Delivery Produce Prescription Program to Address Food Insecurity and Diet Quality in Adults and Children

**DOI:** 10.3390/nu14102006

**Published:** 2022-05-10

**Authors:** Laura Fischer, Nia Bodrick, Eleanor R. Mackey, Anthony McClenny, Wayde Dazelle, Kristy McCarron, Tessa Mork, Nicole Farmer, Matthew Haemer, Kofi Essel

**Affiliations:** 1Department of General and Community Pediatrics, Children’s National Hospital, Washington, DC 20010, USA; lfischer@childrensnational.org (L.F.); nbodrick@childrensnational.org (N.B.); 2School of Medicine and Health Sciences, George Washington University, Washington, DC 20037, USA; emackey@childrensnational.org (E.R.M.); wdazelle@gwu.edu (W.D.); 3Center for Translational Research, Children’s National Hospital, Washington, DC 20010, USA; 4Children’s National Hospital, Washington, DC 20010, USA; amcclenny@childrensnational.org; 5YMCA of Metropolitan Washington, Washington, DC 20009, USA; kristy.mccarron@ymcadc.org (K.M.); tessa.mork@ymcadc.org (T.M.); 6National Institutes of Health Clinical Center, Translational Biobehavioral and Health Disparities Branch, Bethesda, MD 20892, USA; nicole.farmer@nih.gov; 7Section of Nutrition, Department of Pediatrics, University of Colorado Anschutz Medical Campus, Aurora, CO 80121, USA; matthew.haemer@cuanschutz.edu

**Keywords:** food insecurity, diet-related disease, nutrition education, produce prescription, eating behavior

## Abstract

Produce prescription programs aim to improve food insecurity (FI) and nutrition but their effectiveness is unclear. We conducted a pilot study to demonstrate the feasibility and explore the potential impact of a family-based, home-delivery produce prescription and nutrition education program. We measured enrollment, satisfaction, participation, and retention as measure of feasibility. Adult participants answered pre-post self-report questionnaires assessing FI, child and adult fruit and vegetable intake, and culinary literacy and self-efficacy. To understand participants’ lived experiences, qualitative interviews were conducted at the 6-month time point. Twenty-five families were enrolled. Feasibility measures indicate participants were generally satisfied with the program but there were important barriers to participation. Qualitative data revealed themes around reduced food hardship, healthy eating, budget flexibility, and family bonding. Fruit and vegetable consumption increased in a small subgroup of children, but post-intervention intake remained below recommended levels, particularly for vegetables. FI scores were not significantly different post-intervention, but qualitative findings indicated improved access and reliability of food. This is the first intervention of its kind to be evaluated for feasibility and our results suggest the intervention is well-received and supportive. However, further study, with a larger sample size, is needed to understand factors influencing participation and assess effectiveness.

## 1. Introduction

Almost 15% of US households with children reported experiencing food insecurity (FI) in 2020 [1]. The COVID-19 pandemic exacerbated FI and impacted specific subpopulations disproportionately, including households with children and those led by a single parent, in addition to low-income and racial/ethnic minoritized (Black, Hispanic, and Indigenous) households [1]. Despite a number of initiatives such as bolstering child tax credits, developing the Pandemic Electronic Benefit Transfer (P-EBT) program, strengthening the pre-existing Federal nutrition assistance programs (i.e., Supplemental Nutrition Assistance Program (SNAP, formerly food stamps) and Special Supplemental Nutrition Program for Women, Infants, and Children (WIC)), and even strengthening the charitable food system [1], FI remains a public health problem. Addressing FI should be a public health priority because it is associated with poor diet quality [2] and health and social outcomes across the lifespan, including increased cardiometabolic disease risk [3], reduced quality of life [4], social stressors [5,6], and poor mental health [7], including symptoms of depression and anxiety [8,9]. Recently, FI has become more of a national priority, with US President Biden’s administration committing USD 5 billion to domestic FI solutions that are science-based and improve access to food and healthy eating [10].

Associated with FI, eating a diet low in fruits and vegetables places individuals at increased risk of heart disease, obesity, and certain cancers [11,12], all of which increase the risk of COVID-19-related morbidity and mortality [13]. The US Dietary Guidelines recommend children consume 0.5 to 2 cups of fruit and 0.6 to 2.5 cups of vegetables daily and young adult women should consume 1.5 to 2 cups of fruit and 2 to 3 cups of vegetables daily [14]. Unfortunately, nearly 90–95% of American adults and children do not consume the recommended amounts of daily vegetables or fruit, with the lowest intake in those living in poverty [15,16]. Moreover, low intake of fruits and vegetables is common in historically socially and economically marginalized populations struggling with social determinants of health such as poverty, housing, and FI [17,18,19]. Research suggests barriers to fruit and vegetable consumption in households experiencing FI are varied and complex [20] and include cost, desirability, accessibility, and limited culinary knowledge and efficacy [18,21,22,23,24].

Produce prescription programs have been implemented in primary care settings to address FI and diet quality and many targeting adults have been evaluated for feasibility and impact [25,26]. Few programs specifically addressing families with young children have been evaluated and all, except one [27], have used a client-choice, external-to-the-home model of providing supplemental produce resources (e.g., farmer’s market or food pantry vouchers) [28,29,30,31,32,33,34]. There are very limited data on programs that deliver produce directly to the homes of families with young children. It remains unclear to what extent produce prescription interventions impact FI and healthy eating in families living in under-resourced communities with limited access to healthy food in their community and what is the best way to deliver these interventions. Barriers to healthy eating include the neighborhood food environment [35] and limitations related to transportation and cost [36,37]. Understanding that food access barriers can impact healthy eating, we specifically designed an intervention that attempts to bypass access barriers and offer produce directly to the participants in a home-delivery model.

The Children’s National Hospital Family Lifestyle Program’s (FLiP) Home-Delivery Produce Prescription Initiative (FLiPRx) aims to address FI and diet-related disease risk in families with young children living in under-resourced settings in Washington, DC and its surrounding metropolitan areas. Many families in Washington, DC experience disproportionately high levels of FI, health inequities, and severely limited access to healthy foods [38]. The FLiPRx program provides fruits and vegetables via home delivery and educational resources that give families culturally tailored knowledge and skills to make healthy choices and encourage long-term healthy eating behavior change. We think this intervention can reduce FI and increase fruit and vegetable intake in adults and children. We conducted a pilot study to demonstrate the feasibility and explore the impact of the FLiPRx program on FI and fruit and vegetable intake and to guide the design of future iterations of the program.

## 2. Materials and Methods

### 2.1. Study Setting and Intervention Design

FLiP is a family-centered, clinical-community collaborative started in 2017 at Children’s National Hospital (CNH) in Washington, DC alongside the Young Men’s Christian Association (YMCA) of Metropolitan Washington, and the American Heart Association (AHA), to better address FI and diet-related chronic disease. This collaborative body, in partnership with 4P Foods, a local food hub, designed the FLiPRx program alongside community stakeholder support and guidance. In November 2020, participants were recruited from two CNH outpatient clinics located in Washington DC’s most historically disadvantaged and under-resourced areas. During well child clinic visits, adults who presented with children aged 0–5 years old were universally screened by clinicians for FI with a 2-question screener (Hunger Vital Sign, HVS) [39]. Adults who screened positive and had their child present for the clinic visit (reference child) were referred to the FLiPRx program by their pediatric clinician. Referred families were further screened for diet-related disease risk factors before being enrolled in the program. We screened a convenience sample of 33 referred families, and we enrolled the first 25 who were eligible.

The intervention consisted of two parts: (1) fresh fruit and vegetable delivery; and (2) virtual nutrition education. For the food delivery, enrolled families received a 12-month supply of bi-weekly (i.e., every two weeks) delivery of fresh produce. The produce delivery was approximately 8 pounds of seasonal, locally sourced, fresh fruits and vegetables (adjustable by request). Families received texts from the delivery team when their box of produce was on its way and again when it was dropped off at the doorstep or preferred location. For the educational component, our virtual evidence-based nutrition education was based on the SNAP-Ed curriculum and was culturally tailored to our target audience. The culturally tailored curriculum prioritized shared norms, beliefs, expectations, and behaviors common in our local predominantly African-American community. We aimed to establish racial concordance [40] and highlighted the lived experience of local community members in all educational material. Families shared recipes, recommendations, and cooking strategies amongst peers, aligned with the curriculum. Lastly, our adaptations were supported by participants and our community advisory board. The nutrition education offered approximately 24 h of content in the form of monthly virtual cooking classes (a 1 h demonstration and question/answer session with a chef and registered dietitian), bi-weekly brief video-based education (“FLiP Tip” videos provide an evidence-based nutrition tip from a local African-American community physician), brief recipe videos (demonstrating a nutritious recipe with produce provided in the delivery), and recipe and skill-building instruction cards included in every produce delivery (written recipes utilizing the produce offered and culinary skill instructions). Every other Monday, enrolled adult participants received a text message with the “FLiP Tip” video link. Two days before the monthly class and on the day of the class, adults received a reminder text with the Zoom invitation link. At the end of the 12-month program, we offered participants additional FI and SNAP-Ed nutrition education resources through the YMCA and our FLiP Patient Navigators who helped ensure families had access to federal and local nutrition programs.

Lastly, the design of our program aimed to address multiple barriers to healthy eating associated with FI. The large amount of local produce provided aimed to address not only the quantity of food available in the household but the quality, thereby supporting the notion of enhancing the nutrition security of families and supporting the economic investment in local community. The home delivery of food ensured reliability in an urban community where transportation and full-service grocery stores are limited. In addition, we aimed to increase nutrition and culinary knowledge using culturally tailored education delivered by trusted community-based partners and clinicians.

### 2.2. Overview of Study Design

We conducted a pilot mixed-methods longitudinal behavioral intervention to assess the FLiPRx program design and feasibility and to explore impact. This was a quasi-experimental approach without a control, using a pre-post quantitative evaluation and qualitative interviews. We chose the mixed-method design because qualitative assessments are a valuable tool to gain a better understanding of the lived experiences of participants and to put the quantitative findings into perspective.

Quantitative measures used to assess feasibility were rate of program enrollment, retention, satisfaction survey response rate, and participant-reported satisfaction with program content. Quantitative measures to assess impact were pre-post questionnaires with questions derived from standardized measures of FI, eating behavior, home environment, and culinary and health self-efficacy. Adult participants completed the self-report survey at baseline and the same survey again at the 12-month conclusion of the program. Surveys were conducted online using REDCap electronic data capture tools [41]. Qualitative measures used to assess the program feasibility and satisfaction and explore impact were one-on-one interviews. We conducted qualitative interviews at the mid-program point to assess participants’ thoughts and feelings about the program design and its impact on behaviors and to contextualize the quantitative findings. 

Inclusion criteria for eligibility were: (1) positive FI status: families must screen positive for being “at risk” for FI by HVS screen, (2) reference child aged 0–5 years old, (3) positive diet-related chronic disease risk factor, and (4) willingness to complete monthly surveys. Chronic disease risk was defined for reference children younger than 2 years old as having any of the following criteria: abnormal weight gain (>2 standard deviation (SD) increase in growth on either weight-for-age or weight-for-length based on The World Health Organization (WHO) growth charts within the last 1 year), overweight status (weight-for-length ≥97.7% WHO growth charts), or adult caretaker has reported a diet-related chronic disease such as hypertension or high blood pressure (BP), pre-diabetes, diabetes (DM), heart disease, stroke, or obesity. Disease risk was defined for reference children older than 2 years old as any of the following: abnormal weight gain (>2 SD increase in body mass index, BMI within the last 1 year), overweight or having obesity (BMI >85th%), elevated BP (>90th%), pre-diabetes, or DM. After being screened for eligibility, adult caretakers gave verbal consent to participate in the program, which was approved by the Children’s National Hospital Institutional Review Board.

### 2.3. Measures of Feasibility (Enrollment, Survey Response, Satisfaction, Participation, and Retention)

We assessed program feasibility by monitoring enrollment, response to satisfaction surveys, participant-reported satisfaction, monthly class attendance and video views, and retention. Rate of enrollment was determined by the number of eligible families who agreed to participate divided by the number of families screened.

The rate of response to satisfaction surveys was determined by the number of completed satisfaction surveys divided by the number of enrolled participants during the response period.

Satisfaction was assessed by self-report surveys, which included questions about satisfaction with the produce offerings and educational components, and utilization of the produce provided. Responses to satisfaction questions ranged from 1 (not at all satisfied/dislike) to 5 (completely satisfied/very much like) and a sliding scale to indicate percentage of produce utilized. Invitations to satisfaction surveys were sent to all enrolled adult participants by text and email and three automatic email reminders were sent if needed. Satisfaction surveys were sent bi-weekly during the first 6 months of the program, then were sent monthly during the last 6 months of the program (a total of 17 satisfaction surveys were sent to enrolled adult participants over the course of the 12-month study). We asked all enrolled participants to complete the satisfaction surveys but there were no specific disenrollment criteria related to survey completion.

Participation was assessed by monthly virtual class attendance as observed by program staff present at each class session and self-report virtual class attendance and video views at post-intervention.

Six- and twelve-month retention was determined by the number of enrolled families (receiving produce, education, and survey invitations) at the 6-month and 12-month time points divided by the total number of originally enrolled participants.

### 2.4. Baseline and Post-Intervention Questionnaire

#### 2.4.1. Demographics and Anthropometrics

Participants were asked their age, gender, marital status, household size, income, education level, employment status, federal supplemental nutrition program participation, child age, height, and weight to calculate BMI and BMI category at baseline.

#### 2.4.2. Food Insecurity (FI) Variables

We used the six-item (12-month) USDA screener [42] to obtain an FI score. Answers that endorsed circumstances related to FI were given a score of 1, answers that did not endorse FI were given a score of 0. The FI score was the sum of the six questions and was used to create FI category variables (with higher scores indicating higher FI): very high FI (5–6), high FI (2–4), and marginal FI (0–1). FI was measured at baseline and post-intervention.

#### 2.4.3. Fruit and Vegetable Consumption

We asked adults to answer questions derived from food frequency questionnaires about their own eating habits and their child’s eating habits at baseline and post-intervention. For adult intake, we asked about frequency of daily intake in the past week (“About how many cups of fruit (NOT 100% pure fruit juice) did you eat each day?” and “About how many cups of vegetables (NOT 100% vegetable juice) did you eat each day?”) and provided guidance about serving size (e.g., “One cup of fruit is approximately 1 cup of raw or cooked fruits”). Daily frequency categories were converted into a continuous numerical variable (intake score) by assigning an arbitrary numerical value (1–7) to each categorical serving size option (“none”, “½ cup or less”, “½–1 cup”, “1–2 cups”, “2–3 cups”, “3–4 cups”, “4 cups or more”).

For child intake, adults answered questions about frequency and portion size of fruit and vegetable consumption (excluding potatoes) adapted from the National Cancer Institute’s Eating at America’s Table Study Quick Food Scan [43] and the corresponding scoring algorithm [44]. Adults only answered these questions if they reported their child was eating solid food at the time of the survey. Per other longitudinal produce prescription programs [28,29,32], we might expect to see an increase in fruit and vegetables of 0.1 to 0.3 cups. Questions were asked about fruits and vegetables separately (e.g., “How many days per week does your child have fresh fruit” and “How much on those days”) and descriptions of fruit and vegetable types and portion sizes were offered (e.g., “for example, apple with skin” and “½ cup equals one small fruit”). To obtain daily cup equivalents, we multiplied weekly frequency of intake by quantity consumed and divided by 7. This is the equation used to find fruit and vegetable cup equivalents:Daily cup equivalents = (Frequency of Intake × Quantity Consumed) ÷ 7.(1)

#### 2.4.4. Modified Family and Nutrition and Physical Activity (FNPA)

The FNPA screening tool is designed to evaluate the home environment and family practices that may contribute to a child becoming overweight. It has been shown to have good internal consistency and utility for predicting children’s risk for becoming overweight [45]. We used the 11 questions that were most relevant to our purpose. Responses ranged from 1 (never/almost never) to 4 (very often/always) and were averaged to create an individual’s modified FNPA scale score.

#### 2.4.5. Modified Perceived Health Competence Scale (PHCS)

The PHCS is used to measure a participant’s perceived ability to manage their own health outcomes [46]. We asked 8 questions that were the most relevant to our purpose. Responses to these questions ranged from 1 (strongly disagree) to 5 (strongly agree). Responses were averaged to create an individual’s modified PHCS score.

#### 2.4.6. Modified Food Resource Management (FRM)

These questions are specific to participant’s shopping behaviors and self-confidence in buying and using healthy foods. Four questions were derived from the Cooking Matters Questionnaire [47]. Responses to these questions ranged from 1 (strongly disagree) to 5 (strongly agree). Responses were averaged to create an individual’s modified FRM score.

### 2.5. Qualitative Assessment

To conduct the qualitative assessment, we developed an interview template with the following research objectives in mind: to assess the program feasibility and satisfaction with the produce and nutrition education offerings, and explore the impact of the program on FI and eating behaviors. The instrument contained a set of open-ended questions, accompanied by probes, which provided structure to the discussion, while allowing the flexibility needed to allow for spontaneity and candor. The instrument was designed to elicit feedback on the following topics: (1) opinions of the produce delivery and the nutrition education content, (2) benefits experienced from the produce delivery and nutrition education offerings, (3) influence of program on motivation to prepare and eat fruits and vegetables in the family, (4) impact of program on perceived FI, and (5) suggestions for program improvements.

To schedule the interviews, the study team attempted to contact the original 25 participants by phone, approximately 6 months into the program. The research team was unable to contact 10 participants after multiple attempts due to the phone being disconnected, or no response to email/phone call. Thus, of the 25 participants, 15 were reached by phone and invited to participate in the interview, for which they would receive a USD 20 incentive. Twelve adults agreed and completed the interview. One-on-one interviews were conducted by LF securely online using Zoom and were recorded. One of our two research assistants (OS, KM) joined each interview as a note taker. After interviews were completed and recordings were transcribed, the research team used a thematic analysis framework manually to analyze the qualitative data. Members of the research team (LF, NB, KE, OS, and KM) read the participant responses to become deeply familiar with the content. After this initial review, members of the research team (LF, KE, OS, and KM) met as a group to review an initial set of “themes” or commonalities generated from participant statements. Inductive coding techniques were used to ensure preliminary theme development was identified by the content of the data rather than by preconceived models or concepts. Themes were considered overarching topics while subthemes were more specific topics that fell under each theme. These were placed in a codebook with example quotes that represented each theme and subtheme. Members of the research team (OS, KM) then used this codebook to independently analyze and code each interview. Codes were reviewed frequently within the research team; after saturation was reached, emerging themes and subthemes were finalized. Any discrepancies were resolved through group consensus (LF, NB, KE, OS, and KM).

### 2.6. Statistical Analysis

Univariate descriptive statistics (e.g., frequencies and means) were used to describe the sample demographics and feasibility and outcome variables, and Shapiro–Wilks test was used to explore normality of the data. This study was not powered to show statistical significance; however, for the sake of exploration, we conducted exploratory bivariate statistical analyses. Student’s *t*-test was used to examine pre-post group differences in continuous variables (FI score and child fruit and vegetable intake) and chi-square was used to evaluate pre-post differences in categorical variables (FI category and adult fruit and vegetable intake frequency). To explore the relationship between FI and diet, we used Pearson correlation to evaluate the relationship between FI score and fruit and vegetable intake in adults (intake score) and children (cup equivalents). Alpha was set at 0.05. Analysis was conducted using GraphPad Prism 9.3.1 statistical software (San Diego, CA, USA).

## 3. Results

### 3.1. Program Delivery

Over the course of the 12-month program, approximately 4000 pounds of produce was delivered to participants’ homes along with 72 recipe and skill cards. Twenty-six “FLiP Tip” physician videos were sent by text message to enrolled participants and 12 monthly virtual cooking classes were held.

### 3.2. Feasibility 

#### 3.2.1. Rate of Enrollment

The study population was recruited by convenience sampling from two Children’s National Hospital outpatient clinics and is representative of families living in under-resourced areas with a predominantly African-American demographic in Washington, DC, USA. In December 2020, 33 families were referred and screened by phone for diet-related disease and 25 eligible families were enrolled in the program. Therefore, rate of enrollment was 75.8% (25/33). Screening, recruitment, and enrollment were completed in 3 weeks.

#### 3.2.2. Survey Response 

The rate of satisfaction survey completion was lower than expected (<50% response rate) during the first 6 months of the program. To encourage greater survey participation, we revised our survey strategy to include a USD 5 incentive for completing a survey; multiple telephone, text, and email invitations, reminders, and follow-ups were sent; and we reduced frequency of surveys from bi-weekly to monthly. There were no other changes to the study. During the next 6 months, satisfaction survey response rate increased to an average of 77% over the course of the remaining 6 months.

#### 3.2.3. Satisfaction 

Satisfaction survey responses revealed that most families were very satisfied with the program, indicating they liked the produce and the educational components. On average, participants reported 77.5% of the produce was used or frozen for future use. Regarding produce satisfaction, 80% of respondents were very or completely satisfied with produce variety and 80.5% said they tried a new food when one was delivered (e.g., bok choy and squash were specifically listed as examples of new foods participants tried for the first time). Regarding recipe liking and accessibility, 55.9% said they made the recipe (exactly or modified), only 15.9% said they did not have the tools or ingredients to make the recipe, and 11.8% said they did not like the recipe. Regarding the video-based nutrition education content, 76.5% indicated they somewhat or very much liked the FLiP Tip videos. 

#### 3.2.4. Participation 

Post-intervention self-report survey questions asked participants what percentage of videos they viewed (out of 26) and what percentage of classes they attended (out of 12). The average percentage of videos viewed per participant was 66.8% +/− 28.1% and the average percent of classes attended per participant was 63.1% +/− 22.7%. Observed monthly attendance of enrolled families was between 4% (1 out of 25, month 1) and 27% (5 out of 18, month 12).

#### 3.2.5. Retention

At the 6-month time point, we contacted all 25 enrolled participants to describe our revised survey strategy and to ask about any barriers to their continued participation in the program. At this point, 7 were disenrolled in the program and their produce delivery and nutrition education ended. Of the 7 who ended participation at the 6-month time point, 2 were lost to follow-up (unable to reach) and 5 decided to end their participation voluntarily, largely citing time and responsibility constraints. Retention at 6 months was 72% (18/25) and attrition was 28% (7/25). At 12 months (program completion), 15 families completed the post-intervention survey; therefore, 12-month retention was 60% (15/25) and 12-month attrition was 40% (10/25).

### 3.3. Baseline Demographics and Anthropometrics

All adult participants were African-American females with a mean age of 29.9 +/− 5.8 years old. Median household size was 1 adult and 3 children. Unemployment rate of adult participants was 40%, 48% had a high school diploma, and 40% were making less than USD 10,000 per year. A majority of families have used either SNAP (64%), TANF (56%), and/or WIC (56%). Because diet-related disease risk in parents was an inclusion criterion, not surprisingly, adult participants had a high prevalence of obesity (56%) and high BP (40%). Average adult BMI was 33.7 +/− 1.9 kg/m^2^. See Table 1.

### 3.4. Food Insecurity (FI)

In general, the average FI score decreased from baseline to post-intervention, representing a perceived improvement in FI. The average baseline FI score was 2.84 +/− 1.9 and the average post-intervention FI score was 2.4 +/− 1.5. This pre-post difference was not statistically significant (*t* = 0.74, *p* = 0.46) but represents a trend in the expected direction, indicating an improvement in perceived FI. We also examined the pre-post change in FI category distribution. Fewer adult participants ranked in the very high FI category post-intervention. At baseline, 32% reported experiencing very high FI vs. 7% post-intervention. These pre-post differences were not statistically significant (chi-square 4.55, *p* = 0.10) but indicate a trend toward a difference in the expected direction (see Figure 1).

### 3.5. Food Frequency and Feeding Habits 

#### 3.5.1. Child under Age 1 Year Old

There were 11 adult caretakers of children under the age of 1 year old enrolled in the study. The majority (7/11, 64%) indicated they were giving their child formula, of which 5 were exclusively formula-fed. Among those using formula, the predominant type was dairy-based (71.4%) vs. soy-based (28.6%). Four adults reported giving breastmilk, of which three where exclusively breastfeeding. There were 3 (27%) adults who reported giving solid food (so they were included in our analysis of fruit and vegetable intake). In general, adults reported starting solid foods at 1–3 months (2, 12%), 4–6 months (9, 53%), 7–9 months (4, 24%), or 10–12 months (2, 12%). However, this question did not specifically ask adults to report solely on this practice for the reference child; therefore, adults may be reporting when they historically started solid food for other children in the home. 

#### 3.5.2. Over Age 1 Year Old, (or Age < 1 Who Were Fed Solid Food)

At baseline, there were 17 adult participants who answered a food frequency questionnaire for children. At post-intervention, there were 14 (only 10 overlap because over the course of the 12-month study, 4 children progressed from not eating solid food at baseline to eating solid food, hence only post-intervention dietary data are available). The average parent-reported baseline and post-intervention daily intake of fruit was somewhat close to the amount recommended by the USDA Dietary Guidelines (0.5–2 cup) and close to the national average for young children (0.8 +/− 0.04 cups) [48]. Vegetable intake was much lower than the recommended amounts (0.6−2.5 cups) [14] and lower than the national average for young children (0.7 +/− 0.03 cups) [48]. However, there was a general increase in fruit and vegetable consumption post-intervention. The pre-post average daily cup equivalent of fruit increased from 0.73 +/− 0.50 cup equivalents at baseline to 1.1 +/− 0.71 cup equivalents post-intervention, representing a 34% increase and an absolute change of 0.38 cup equivalents. This pre-post difference was trending toward statistical significance (*t* = 1.7, *p* = 0.09). The average daily cup equivalent of vegetables increased from 0.34 +/− 0.27 at baseline to 0.52 +/− 0.32 post-intervention, representing a 34% increase and an absolute change of 0.18 cup equivalents. This pre-post difference was trending toward statistical significance (*t* = 1.7, *p* = 0.10). See Figure 2.

We wanted to know if children were meeting the recommended daily intakes at baseline and post-intervention. We defined “adequacy” of daily intake as the low end of the recommended range for age groups, that is, 0.5 cups of fruit for children under 1 year old and 1 cup of fruit for children over 1 year old, 0.6 cups of vegetables for children under 1 year old, and 1 cup of vegetables for children over 1 year old. At baseline, very few children met this conservative recommended intake for vegetables (6%), but a larger proportion of children met the fruit recommendation (41%) and only 1 (5%) met both fruit and vegetable recommendations. At post-intervention, the proportion of children who met the vegetable and fruit recommendations increased, but “adequate” vegetable intake was still low (21%), while more than half met the fruit recommendation (57%) and 20% met both fruit and vegetable recommendations.

If we only look at the subset of 10 participants with complete pre-post data, we can use the more sensitive statistical test (paired *t*-test) to test the significance of the pre-post differences in average fruit and vegetable intake. The change in fruit and vegetable intake was greater in the subgroup of participants with complete pre-post data but the change in vegetable intake was smaller. There was a 43% increase in average daily fruit intake (0.76 +/− 0.53 to 1.3 +/− 0.72), and only a 29% increase in average daily vegetable intake (0.44 =/− 0.30 to 0.62 +/− 0.32). The difference in average fruit consumption was statistically significant (*t* = 2.8, *p* = 0.02) and the difference in average vegetable intake was not (*t* = 1.4, *p* = 0.21).

#### 3.5.3. Adults

Adult self-reported that baseline and post-intervention consumption of both fruits and vegetables was generally lower than the recommended amounts (at minimum 1.5 cups of fruit and 2 cups of vegetables) [14]. At baseline, only 32% reported having 1 or more cups of fruit per day and only 20% reported having 2 or more cups of vegetables per day. However, frequency of daily intake of both fruits and vegetables at post-intervention increased compared to baseline frequencies. At post-intervention, 40% reported 1 or more cups of fruit and 40% reported 2 or more cups of vegetables. Because some frequency categories were less than 5, to conduct statistical analysis, we collapsed frequency response options into three categories: <½ cup, ½ cup to 1 cup, and >1 cup for fruit and <1 cup, 1–2 cups, and >2 cups for vegetables. The pre-post difference in frequency of vegetable intake was trending toward statistical significance (chi-square 3.31, *p* = 0.07) but the difference in fruit intake was not statistically significant (chi-square 1.0, *p* = 0.32). See Figure 3.

### 3.6. Exploring the Relationship between FI and Fruit and Vegetable Intake

We examined the quantitative relationship between FI score and fruit and vegetable consumption using Pearson correlation. To conduct correlation analysis, we created a continuous numerical variable for adult fruit and vegetable intake by assigning an arbitrary numerical value (1–7) to each categorical serving size option (“none”, “½ cup or less”, “½–1 cup”, “1–2 cups”, “2–3 cups”, “3–4 cups”, “4 cups or more”). In adults, we found an inverse relationship between baseline FI score and baseline fruit (R = −0.59, *p* = 0.002) and vegetable (R = −0.35, *p* = 0.09) intake. As FI score increased, fruit and vegetable consumption decreased. The relationship was statistically significant for fruit but was only trending toward being statistically significant for vegetable intake. The relationship between adult FI score and fruit and vegetable intake did not persist at the post-intervention time point. In children, there was a similar general trend, a negative correlation between baseline FI score and baseline fruit (R = −0.37, *p* = 0.14) and vegetable (R = −0.09, *p* = 0.72) intake, but none of the correlations were significant.

### 3.7. Home Environment, Self-Efficacy, and Food Resource Management

There was no significant pre-post difference in responses to individual questions or the average sum scores. Baseline average sum scores were relatively high and therefore we may have experienced a ceiling effect, whereby there was not a lot of room for improvement in these scores. The average baseline score for FNPA was 3.0 +/− 0.41, PHCS was 3.6 +/− 0.67, and FRM was 4.4 +/− 0.14.

### 3.8. Comparing Finishers and Non-Finishers

There were 15 participants who finished the study and 10 participants who did not finish the study (non-finishers). We wanted to compare baseline characteristics of participants who finished and who did not finish. More non-finishers reported income greater than USD 25,000 per year (60%) vs. finishers (6.7%). The average baseline FI score was lower in non-finishers vs. finishers (2.4 +/− 1.7 vs. 3.1 +/− 2.1) and there were fewer non-finishers with baseline “very high” FI vs. finishers (20% vs. 40%), indicating non-finishers may have had lower perceived FI and higher income, but we cannot say for certain, as these differences were not statistically significant because the sample size was very small. Baseline scores for modified FRM, PHCS, and FNPA did not differ in non-finishers vs. finishers. Baseline child and adult fruit and vegetable intake did not differ by parent finish status. Although we cannot make any conclusions about the differences in finishers and non-finishers at baseline, it is worth exploring in subsequent studies.

### 3.9. Qualitative Interviews

Four preliminary themes were derived from qualitative interviews: (1) reduced food hardship, (2) support for family-driven behavioral change, (3) increased economic flexibility, and (4) opportunities for family bonding.

Within each of these main themes were subthemes. The subtheme for theme 1 was improved food accessibility and reliability. For example, participants described how having a consistent source of food, exposure to a wider variety of food, and direct delivery to their home improved the quantity and quality of food available in the home. This supports our quantitative results described in Section 3.4, which suggest a trend toward a reduction in FI severity category.

Under theme 2, we specifically identified eating and cooking behaviors as those that were impacted. As an example, participants described trying new foods and learning new kitchen and food preparation skills as a result of engaging in the program. However, we did not find this reflected in our quantitative assessment of culinary self-efficacy as described in Section 3.7. Qualitative results indicate the greatest impact of the program may not be directly on the adult but may be on facilitating the transfer of this knowledge from parent to child. “the children were really really engaged … the more engaged the children are, the more they want to taste the food and like to try the food for the first time.”

Under theme 3, participants described how the program eased some of the impact on household economics and financial strain at the end of the month when SNAP benefits are low. More than 60% of participants in the intervention were using SNAP, and by providing food, families were able to use their money on other non-food expenses. Participants describe using money on non-food expenses such as bills, gasoline, other grocery foods, gifts, entertainment, and dining outside the home. Under theme 4, participants indicated they felt a greater sense of family togetherness around food preparation and eating and that the program specifically encouraged child engagement in the cooking process. See Table 2 for example quotes.

## 4. Discussion

### 4.1. Summary of Major Findings

The aims of this study were to examine the feasibility of implementing a home-delivery produce prescription and virtual nutrition education intervention and to explore the impact of the intervention on FI and fruit and vegetable intake in children and adults. We found enrollment of the target population was feasible and the program was well-received; however, early measures of program participation indicated there were unanticipated barriers to participation. In both quantitative and qualitative assessments, adult participants reported positive thoughts and feelings about the fruit and vegetable delivery and an increase in intake of fruit and vegetables in a subgroup of children. FI scores generally went down but the change was not significant. Semi-structured interviews revealed positive impacts of the program on food access, family fruit and vegetable consumption, and the family budget, and shared enthusiasm for the preparation and use of fruits and vegetables at home. This is promising evidence to support our exploratory hypothesis that the FLiPRx intervention can impact FI and fruit and vegetable intake, but we are not able to draw conclusions about the causality of this outcome. More work is needed to understand barriers and facilitators of program participation and the impact of the program on nutrition literacy, eating and purchasing behaviors, and FI.

### 4.2. Fruit and Vegetable Intake Improved in a Subgroup of Participants in a Similar Magnitude to Other Produce Prescription Studies but Consumption Was Still Below Recommended Levels

Although the study was not specifically powered to test for a statistically significant increase in fruit and vegetable intake, we explored the impact of the program on eating behavior change. We found a post-intervention increase in fruit intake in a subgroup of children, although intake was still generally lower than the federally recommended amounts. The proportion of adult participants that met intake recommendations somewhat increased from baseline to post-intervention by 20% (32% to 40%) and by 50% (20% to 40%) for fruit and vegetables, respectively, but this was not statistically significant. Child average intake of fruit and vegetables generally increased post-intervention (by 0.38 cup equivalents and 0.18 cup equivalents, respectively); however, the consumption at baseline and post-intervention was below recommended levels, with vegetable intake farther below the recommendation than fruit intake. This is consistent with data reporting an innate predilection towards more sweet taste in children [49]. The proportion of child participants that met the daily cup recommendation increased from baseline to post-intervention by 28% (41% to 57%) and 71% (6% to 21%) for fruit and vegetables, respectively. This is encouraging because, nationally, child vegetable intake has remained steadily low since 2003 [16] and can be difficult to change, but is possible with multiple repeat exposures [50]. Our qualitative data suggest a relationship between family food involvement and willingness to try fruits and vegetables in children. There is other evidence that outreach efforts focused on family food involvement in early childhood may improve children’s dietary habits [51].

Few others have reported longitudinal quantitative assessments of child fruit and vegetable intake in response to a primary-care-based produce prescription program. We identified three relevant publications with quantitative fruit and vegetable results in children. A produce prescription program in Flint, MI with children aged 2–18 years old showed increased fruit intake (by 0.19 cups) and no change in vegetable intake at 6 months evaluation time point [29], but reduced fruit intake and increased vegetable intake (by 0.23 cups) at 12 months evaluation time point [28]. Another produce prescription program through Federally Qualified Health Centers described increased fruit and vegetable consumption by 0.1 to 0.3 cups [32]. The magnitude of change in child fruit and vegetable intake reported by other produce prescription programs is similar to our results (0.3 cup equivalent increase for fruit and 0.18 cup equivalent increase for vegetables).

Having observed lower than optimal fruit and vegetable intake post-intervention, we recognize there is room for even greater improvement in consumption in participants of this program. However, there was no explicit intake goal in this pilot study; we simply wanted to know if pre-post intake would differ by any amount. A goal of future work could be to help participants achieve a specific intake goal in line with the federal recommendations for a healthy diet. Moreover, it is a well-known phenomenon that any behavior change takes time and consistent effort. Our qualitative findings suggest increased familiarity and involvement in the family food preparation process may be a key factor in promoting fruit and vegetable consumption in children. The short duration of this study could mean we have not yet seen the eventual increase that will take place with continued exposure to fruits and vegetables, given participants continue purchasing and involving children in the preparation of fruits and vegetables after they finish the program. In addition, because of the rich descriptions of fruit and vegetable intake in the qualitative evidence, it is possible that the methodology for collecting fruit and vegetable intake measures was not adequate to show the families’ lived experiences. Although it can be a burdensome effort, future work should use more rigorous dietary assessment methods, such as the repeated 24 h recall [52].

Another consideration regarding the magnitude of change we observed in fruit and vegetable consumption is that there are many complex drivers to intake; among them are cost and accessibility, which this study specifically addressed, but also taste preference and familiarity [20] and the hierarchy of needs [53], which were not specifically addressed within the scope of this study. A multi-component intervention that addresses the complex fruit and vegetable intake factors will have the greatest opportunity for success. FLiPRx was designed to specifically address cost (free), accessibility (home delivery), and familiarity (consistent exposure over 12 months). It is difficult to address taste and preference but these are modifiable factors [50]. Future work should explore the impact of produce prescription interventions on these factors and how they may modulate change in fruit and vegetable intake.

### 4.3. FI Scores Were Unchanged but Perceived FI Severity May Be an Important Outcome to Explore as Well as the Relationship between FI and Fruit and Vegetable Intake

In qualitative interviews, we heard from participants that a unique benefit of the FLiPRx program is the stability and increased accessibility that the home-delivery model offers. In qualitative interviews at the six-month mark, participants described the consistency and reliability of the FLiPRx produce delivery as contributing to a decreased perception of FI, particularly because the food was delivered to their home on a consistent basis. From quantitative assessments, we reported fewer participants in the very high FI category post-intervention (7% vs. 32% at baseline) and the average FI score was slightly lower by 15% (2.8 to 2.4), but neither change was statistically significant. Other produce prescription programs have reported improvements in adult perceptions of FI, for example, Aijer reported a 94.1% decrease in the prevalence of FI at the end of the program [31] using the HVS 2-item screener. Saxe-Custack [28] reported a 55% decrease in FI score on 12-month follow-up, using the USDA 6-item screener.

Families experiencing FI often limit the amount, variety, and quality of food in an attempt to save money and to limit food waste [54] and describe barriers to preparing healthy foods, including cost and time [37]. Before the boost in spending on federal nutrition programs during COVID-19, SNAP has typically been considered insufficient to meet the needs of families. For example, families with children will spend the majority of their monthly budget by day 7 and 80% by day 14 [55]. Similarly, the bulk of fruit and vegetable purchases happen within the first week of benefit distribution and purchases of fruits and vegetables decline at the end of the benefit month [56]. In our qualitative interviews we heard participants’ expressed willingness to try new fruits and vegetables was influenced by having a reliable source of fresh produce. We hypothesize that giving families the ability to explore perishable and novel fruits and vegetables without impacting their SNAP dollars or budget increases their likelihood to eat a greater quantity and wider variety of fruits and vegetables. Given this observation, we wanted to explore the relationship between FI score and fruit and vegetable consumption. We found an inverse relationship between baseline FI score and baseline intake; that is, a higher FI score (worse conditions) was correlated with lower fruit and vegetable intake. However, this relationship did not exist at the post-intervention time point and no relationship existed between FI and child fruit and vegetable intake at baseline or post-intervention. These data suggest that: (1) when given easy access to low-/no-cost produce and nutrition information, adults’ fruit and vegetable intake is not correlated with FI status, and (2) adult but not child intake of fruits and vegetables is more immediately impacted by FI status. Regarding adult intake, as far as we know, this is the first family-centered produce prescription program to report the post-intervention relationship between FI status and adult fruit and vegetable intake. Regarding child intake, it is a documented phenomenon that when household food resources are limited, parents prioritize intake in the children over their own intake [57]. Similar to our findings, in a cross-sectional analysis of data from a similar produce prescription program, researchers examining the baseline difference in child fruit and vegetable intake by adult caregiver response status to the Food Attitudes and Behaviors Survey questions found no difference in child fruit and vegetable intake by food access status [58].

The reduction of families in the very high FI status indicates that the extremes of FI may very well be buffered with a reliable source of increased benefits such as produce prescriptions or more robust SNAP dollars. We also reported a theme of greater economic flexibility, and we found this intervention allowed participants to spend money on necessities and luxuries rather than/or in addition to enhancing their regular food purchases. This may lead to unintended consequences such as increasing purchase of food outside the home or snacks, which can potentially have an unintended negative impact on healthy eating behaviors. However, it may also lead to purchasing more high-quality food for the family because of the financial freedom. An exploration of purchasing habits through qualitative and/or receipt analysis should be conducted to better describe short-term and long-term food purchasing habits within produce prescription interventions. Programs that address FI by supplying food resources must also address nutrition education and food resource management support to facilitate healthy food purchasing habits even when additional monetary resources are available. Another study reported that a food prescription program allowed individuals to spend money differently without unintended negative consequences [36], but this was in an elderly population who do not face the same household factors as families with young adult caretakers and children. Future work will look more in-depth at financial and budget factors in FLiPRx participants and how FI score impacts attrition, program participation, and nutrition literacy. 

### 4.4. Program Improvements, Scalability, and Sustainability

We noticed early participation was low. Other produce prescription studies have reported limited engagement of participants [30] and identify many contributing factors, including transportation issues [36]. When asked, participants indicated barriers to participation were cited as time, awareness, competing priorities, and life stressors. Participants cited competing priorities of work, childcare, phone upkeep challenges, housekeeping, or simply forgetting as the reason for not completing satisfaction surveys or attending cooking classes. Based on this feedback, at the 6-month time point, we made changes to our survey strategy. To bolster survey response, we reduced the frequency of surveys (from bi-weekly to monthly), offered a USD 5 incentive, and initiated text and telephone reminder invitations and follow-ups. After these changes were implemented, survey response increased for the remainder of the program. We recognize the challenge of juggling competing priorities for families, especially families with young children, and understand the need for a consistent reminder system in future iterations of this program. We will use a HIPAA-compliant communication platform, which allows automated messaging and allows for scalability and less program staff burden. In addition, we have initiated a “Commitment form” for families so they are able to acknowledge their understanding of every part of the intervention before beginning the program and are encouraged to keep a copy of this for their records.

In terms of improvements to curriculum, participants suggested a useful resource would be a comprehensive guide to the produce they received. Recognizing that individuals learn in multiple ways, including visual, auditory, tactile, etc., to augment our video and live learning sessions, we created a print guide to fruits and vegetables that we will provide to all future participants. This printed book contains an indexed list of all produce that might be delivered during the program and provides tips on how to wash, store, and use the produce, especially for young children. To help families engage with the virtual cooking classes and recipes we provide, we have designed a culinary package to be delivered at enrollment that contains staple ingredients, tools, and spices that families might use to prepare the recipes during the intervention.

Produce prescription programs may play a role in buffering FI and improving diet quality but they alone are not a sustainable solution to tackling the FI needs families face. The goal of our produce prescription program is to provide food and education resources with the hope of promoting sustained behavior change. By providing food and education resources as well as skills to improve self-efficacy, we hope that families begin to utilize their own resources to purchase and prepare more produce once they are finished with the program. Federal nutrition programs and tax credits remain some of the most important resources for addressing FI and diet quality as an essential safety net for households with limited income and financial resources. Produce prescription programs may play a supportive role, to serve as a short-term buffer, yet when participation ends, families must be given access to further resources to continue to address the unmet social need while promoting healthy behaviors. Effective interventions must link participants with other community programs and ensure connection to meaningful federal nutrition programs and tax credits. In this way, the FLiPRx program is prioritizing sustained nutrition security and attempting to address some of the social and socioeconomic barriers to healthy eating. Longer-term follow-up is required to truly understand whether this program has any sustained impact on FI and eating behavior.

The sociocultural context in Washington, DC during our program implementation period may have contributed to unique local barriers to participation. Most notably, our first virtual interactive cooking class was scheduled for January 6th, 2021, a day that is now infamous due to the violent actions that occurred on the nation’s capital down the street from the homes of our participants. Additionally, the city has grappled with difficult and tragic instances related to race relations and systemic racism, exacerbated by the COVID-19 pandemic. These social issues greatly impacted the most vulnerable populations already struggling with community disinvestment, limited opportunities for well-paying jobs, affordable housing, childcare, and education. This cluster of sociocultural disruptions increased the stress already prevalent in the lives of our participants, who were typically single parents, caring for young infants and children with low incomes. As voiced by our participants, these stressors made it particularly difficult for caretakers to make time to attend our structured non-recorded live virtual class. We believe these factors contributed to barriers to participation and future research will need to continue to adapt to families by offering more efficient opportunities for participants to engage with instructors and build skills and knowledge.

Outside of social factors, some of our study design factors may have contributed to lower-than-expected participation. For example, attendance was not formally monitored and was not mandatory to continue receiving produce deliveries. We think participation will improve if attendance is mandatory and if we provide more engagement from program staff earlier and more consistently in the program (i.e., regular automated reminder messages). Additional facilitating factors would be to offer classes at multiple days and times to accommodate all schedules.

### 4.5. Strengths and Limitations

There are strengths and limitations to this study. Strengths include the longitudinal design and use of validated assessment tools. Another strength is the intentional design of our intervention to address multiple barriers around FI, including quantity and quality of food, variety, appeal, reliability, and transportation challenges. The USDA 6-item screener not only measures the presence of FI but detects the depth of FI, and the fruit and vegetable questions assessed total intake by asking about portion sizes in addition to frequency. However, the 6-item screener is limited in exploring FI severity and depth and is unable to identify FI severity at the level of children, instead focusing on the household overall. We experienced quick recruitment and enrollment, which indicates that this program can be seamlessly integrated into clinical systems for easy enrollment from clinical screening. A limitation is that the food frequency tool is a self-report measure with potential for recall bias or differential response bias (i.e., the intervention itself can create differential error in reporting, creating a larger bias in post-intervention responses versus baseline responses). Other limitations include short duration, no control, and a small sample size, limiting the generalizability of our finding and long-term sustainability of the outcomes of this pilot. We also recognize that there is potentially a self-selection bias, whereby people with relatively healthy habits were the ones who agreed to participate in our study. There may also be a social desirability bias, whereby adult caretakers overestimate the amounts of fruits and vegetables they or their children eat to appease the research team. Another source of limitation is attrition; those who did not complete the study may have differed in some factor of interest. For example, there were some notable divergences in income and FI characteristics of finishers and non-finishers, possibly indicating income status, and perceived FI category may play a role in attrition. Another limitation is that we were not able to use web analytics to quantify the degree to which video-based intervention components were utilized by participants; we only have self-report data. This is a change we have implemented in the next iteration of the program. Lastly, this intervention was conducted during the COVID-19 pandemic, and due to a variety of robust federal and local support mechanisms (as described earlier), there are confounding factors we cannot control for, which may have impacted our findings.

## 5. Conclusions

We found the FLiPRx intervention was feasible and well-received by participants, but notable barriers to participation need to be addressed in subsequent iterations. Multipronged interventions that aim to enhance nutrition and culinary literacy and overcome socioeconomic and environmental barriers to healthy eating in historically marginalized and underserved populations may play a role in promoting fruit and vegetable intake in this population. More work is needed to understand the role of this intervention on FI and improving eating habits in adults and children long-term and its impact on health outcomes in addition to cost-effectiveness. Future work must include more robust qualitative assessments to better understand barriers and facilitators to participation, change in fruit and vegetable intake, FI status, and sustainability of behavior change. Additionally, robust quantitative assessment tools must be utilized to better capture longitudinal dietary and spending behavior change. Randomized controlled trials with longer-term follow-ups would provide a clearer understanding of the impact of this program and its effectiveness and help determine whether it is a viable adjunct to support improved diet quality in families who rely on federal nutrition programs. Finally, our intervention is designed to address the social needs and social risk of families through a clinical-community collaborative effort and is not designed to address the deeper social determinants of health that play a major role in overall health. However, future work must continue to develop more effective strategies to work with families and health systems to impact social change through policy and community initiatives.

## Figures and Tables

**Figure 1 nutrients-14-02006-f001:**
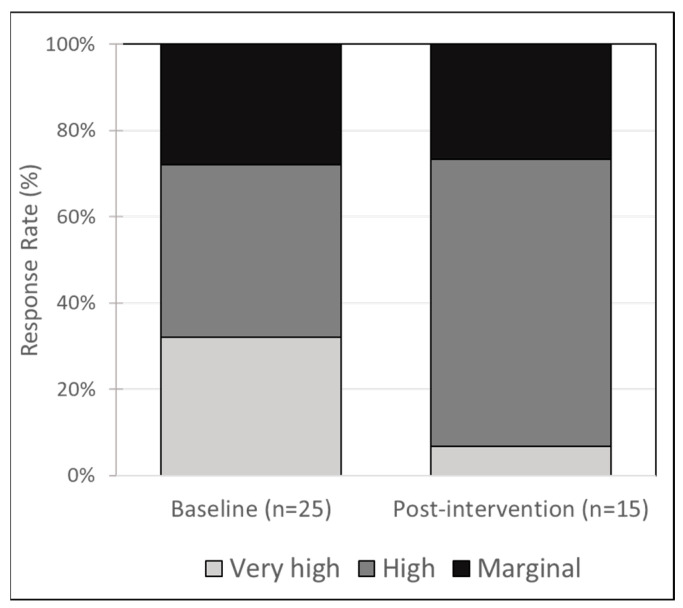
Distribution of food insecurity category at baseline and post-intervention (chi-square test *p* = 0.1).

**Figure 2 nutrients-14-02006-f002:**
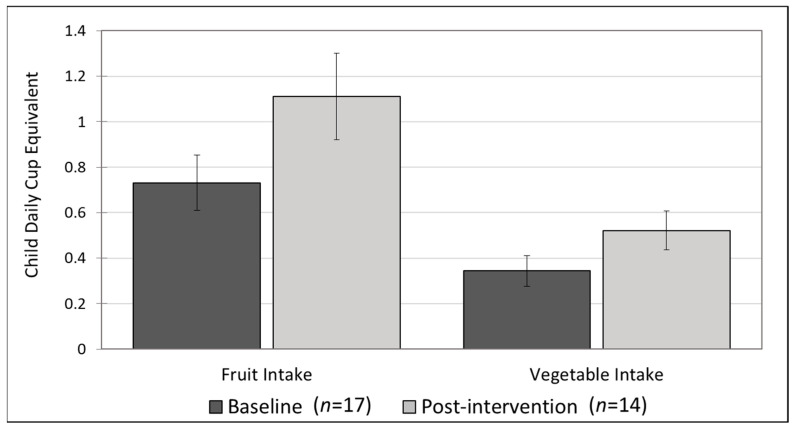
Child fruit and vegetable intake. Average adult-reported child fruit and vegetable intake at baseline (*n* = 17, *t*-test *p* = 0.09) and post-intervention (*n* = 14, *t*-test *p* = 0.1). Bars are group means and error bars are standard error of the mean.

**Figure 3 nutrients-14-02006-f003:**
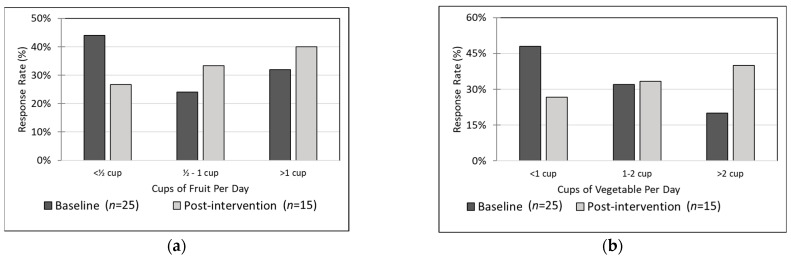
Adult fruit and vegetable intake. Distribution of adult fruit and vegetable intake at baseline (*n* = 25) and post-intervention (*n* = 15). (**a**) Adult fruit intake (chi-square *p* = 0.32). (**b**) Adult vegetable intake (chi-square *p* = 0.07).

**Table 1 nutrients-14-02006-t001:** Baseline demographic data of parent/caretaker.

Variable	Description	*n* (%)
Gender (*n*, %)	Female	25 (100%)
Age (mean, standard deviation, sd)	Age in years	29.9 (5.8)
BMI (kg/m^2^) and BMI category distribution (*n*, %)	BMI (mean, sd)	33.7 (9.4)
	<25	5 (20%)
	25–30	6 (24%)
	30–35	3 (12%)
	>35	11 (44%)
Diagnosis of high BP or DM (*n*, % yes)	High BP	10 (40%)
	DM	2 (8%)
Reference child age group (*n*, %)	0–1 years	11 (44%)
	>1–5 years	14 (56%)
Race (*n*, %)	African-American	25 (100%)
Employment status (*n*, %)	Working full-time	4 (16%)
	Working part-time	6 (24%)
	Going to school or apprenticeship	2 (8%)
	Unemployed	10 (40%)
	Self-employed	1 (4%)
	Prefer not to say	2 (8%)
Level of education (*n,* %)	Less than high school	3 (12%)
	High school diploma or GED	12 (48%)
	Some college	7 (28%)
	College graduate	1 (4%)
	Prefer not to say	2 (8%)
Level of income (*n*, %)	Less than USD 10,000 a year	10 (40%)
	USD 10,001–USD 25,000 a year	3 (12%)
	USD 25,001–USD 50,000 a year	4 (16%)
	Prefer not to say	8 (32%)
Marital status (*n*, %)	Never married/single	18(72%)
	Married or unmarried couple	3 (12%)
	Divorced	2 (8%)
	Prefer not to say	2 (8%)
Household occupants (median)	number of adults	1
	number of children (age 0–17)	3
Governmental support program participation (*n*, %)	FRPS *	8 (32%)
	SNAP	16 (64%)
	SSI *	7 (28%)
	TANF *	14 (56%)
	WIC	14 (56%)
	None	2 (8%)

* FRPS: Free/Reduced Price School Lunch, SSI: Supplemental Security Income, TANF: Temporary Assistance for Needy Families.

**Table 2 nutrients-14-02006-t002:** Qualitative results.

Theme Name	Representative Quote
**Theme 1. Reduced Food Hardship** **Subtheme:** **Improved Reliability** **Improved Food Accessibility**	“They deliver [the FLiPRx produce boxes] so I don’t have to go stand in line, I don’t have to deal with the crowd. It’s just delivery at the front door without me having to order, so that just saves a little time and me trying to get food or going to the grocery store, especially if I’m at work or with the children and don’t have time to take them [or] car’s gone out (Participant #3)” “We love string beans. I was always into my veggies, but I couldn’t afford as much as I get from you all; I couldn’t afford the different varieties” (Participant #17). “[The produce] helps a lot, far as the nutrition, the vegetables and the fruit—I mean different kinds of veggies—for me and my daughter and it just helps us and it saves me money because sometimes I can’t get the veggies and stuff like that that I need because of my finances. Because of the pandemic I lost my job so it helps a lot” (Participant #17). “What if something happens with my SNAP? I can’t just fully depend on that. And now I know that I still have my produce coming from y’all. … It hasn’t been the case yet, but you don’t know what’s gonna happen in the future, so I don’t wanna end the [FLiPRx] program and, you know, something happens with the SNAP, now it’s gotta come out of my pocket. I know I still have the bag coming” (Participant #13).
**Theme 2. Family-driven Behavior Change** **Subtheme:** **Developed healthy eating and cooking behaviors**	‘[I tried] beets. I’ve always thought beets were super disgusting. My mom loves them but we watched one of the [recipe] videos where she was making roasted beets and I was like, ‘oh, okay, I’m gonna try that’ and I did and it was actually really good, so I was like, ‘oh cool!’” (Participant #8) “[The program] has been great not just for the kids. Like for me, I’m consuming more of the fresh produce. It’s done a lot for me and my health. I have been able to stop taking my blood pressure medication so that is a plus. I learned how to give myself the right foods in the right order, to make sure I’m getting enough of the right stuff, and that’s been a big positive” (Participant #19). “[The recipe videos] are pretty cool, it was just like that they’ve really taken the time out and really teaching step by step, you know, especially for those who may not know how to cook or know what to do and I found that pretty cool. […] It’s like they’re getting the hands-on training but it’s virtual and I find that pretty awesome” (Participant #3).
**Theme 3. Economic Flexibility** **Subtheme:** **Stretched the monthly budget**	“[The program] made it so that I was more conscious about what I was purchasing. […] It made me think about meal planning more as opposed to, ‘okay I’m just gonna go and get what I normally get and get out.’ […] If I have a little bit of [FLiP produce], then I can put money that was allotted for [that produce] over to maybe a non-SNAP item or maybe I can get more fruits, more noodles (since that’s what [my son] likes), more meats to go with it to kinda stretch the money a little longer. So it’s actually helped my budget as far as I can now move—my grandmother calls it ‘moving her blocks around’” (Participant #8). “Being the mom in a house with six kids, three adults, you know, sometimes things come up short with my SNAP benefits. I don’t have all the people in my home on my SNAP benefits […] so sometimes we come up short and I have to make those vegetable dishes […] because that’s what we have for us to eat” (Participant #2).
**Theme 4. Enhanced family bonding** **Subtheme:** **Promoted family interactions** **Positive experiences for children around healthy foods and mealtime**	“My two girls love it, especially like cooking, making the little recipes that you guys send in the bag, my children love it. […] They get to prep the food, they get to like, you know, stir the food, make the food, […] they stir the food, sometimes I would show them how to, like, chop the food, like I would guide them with the utensil, things like that” (Participant #4) “That [corn salsa recipe] was pretty good. My sister actually watched the video with me and [we] tried it and it turned out pretty good, it was a little spicy, but it was good” (Participant #10)

## Data Availability

The data are not publicly available due to restrictions (privacy and ethical) but are available on request.

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
