# Peer review of "Feasibility of a Home-Delivery Produce Prescription Program to Address Food Insecurity and Diet Quality in Adults and Children"

_nutrients, 2022, doi:10.3390/nu14102006_

Round 1

Reviewer 1 Report

This paper describes the results from a small pilot study exploring the use of prescription produce program for families with food insecurity. The study addresses an important topic of food insecurity which has been shown to contribute to adult and child health disparities. The study combines produce prescription with education and attempts to assess fruit and vegetable intake over time in families in the intervention. The paper is overall well written, but I had several significant concerns with the manuscript.

Major:

  • The title and conclusion refer to an increase in fruit and vegetable intake when in fact results were not statistically significant. This statement is not accurate. The title should be reframed as well as the conclusion. Focus should be on feasibility which with high attrition is also questionable.
  • The qualitative and mixed methods sections of the paper are not well-defined
  • High Attrition rate of 40% (even if other studies have had higher rates, this is still high). Not clear what criteria was to continue to receive the bi-weekly produce deliveries? Did everyone get 12months or only if you continued to complete the surveys.
  • Pre-post analyses are only done on 10 of the 15 participants who were retained for 12months; no explanation is given as to why there were only 10 overlapping sets of data available. This of course makes the sample size for analysis even smaller.

Minor:

  • There is no information provided on degree to which other intervention components (cooking classes and videos) were utilized by participants.
  • Figures need a title and a p-value and a number of respondents/participants
  • Sample size is small even for a pilot study
  • Six-months data should have been presented à possibility of greater effect at 6-months that then tapers off between 6 and 12months.

Additional points for consideration:

Abstract:

See points below that should be reflected in the abstract.

Introduction:

Knowing that the barriers to food security are multi-factorial, authors should make a case for why focusing on produce delivery is the best first step. In other words, what have other studies shown are barriers to interventions with external to the home program succeeding that this intervention will address?

Methods

  • Describe intervention components (each should then be described and evaluated):
    • Home delivery program
    • 24hours of education: cooking classes, Flip Tip physician videos (bi-weekly), recipe videos, and recipe skill-building.
  • Study design: could be clarified as a pre-post quasi-experimental study design without a control. Evaluated using a concurrent mixed-methods approach.
  • In describing eligibility criteria, authors talk about weight gain of >2 SD but don’t offer a time frame. Weight gain over wat time frame?
  • Consider presenting your quantitative methods followed by your qualitative methods (integrating them can be done but is harder and right now too confusing). If done this way you then have to discuss how you integrated the quant and the qual.
  • Provide rationale for the mixed methods study.
  • Qualitative Methods:
    • Explain how families were chosen for qualitative interviews
    • Clearly describe what the research question was for the qualitative portion of the study
    • How was interview guide designed and piloted?
    • Who conducted the interviews and in what setting?
    • Was incentive provided for the interviews?
    • How many team members coded the transcripts? How were discrepancies resolved?
    • Was an analytic software used?
    • How were themes derived?
    • in talking about BOTH inductive and deductive reasoning please provide examples as often difficult to understand what was done?
  • Define in the methods section what a meaningful change in fruit and vegetable consumption is per prior studies – an increase by x number of cups/servings would signify meaningful change over 1 year.

Results:

  • Explain why only 10 had pre-post intervention data available.
  • Table 2: Qualitative results. Would list subthemes under the themes ( you have enough space)
  • Consider presenting all the quant results and then presenting the qual with the integration element
  • Be consistent in your language to make it easier on the reader: low food insecurity, high food security, low food insecurity score, etc….
  • Hard to justify a correlation analysis with such a small ‘n’
  • In comparing non-finishers to finishers did you have the full 25 person data available, if not provide ‘n’ that was used for that comparison analysis.
  • Would have been important to show participation rates in each of the intervention components: how many cooking classes attended, how many recipe videos watched, how many skill building videos watched. Was participation level associated with success?

Discussion

  • Be careful throughout as results are not statistically significant so very hard to draw conclusions. Especially as there is no control group
  • Would be interesting to see if other produce prescription programs did have a control group and what results there were like?
  • Key factor are the other barriers to food security. Understanding how a program like this does or does not address those barriers is critical – highlighting this is important
  • Understanding why there was such high attrition. I understand other studies may have had higher attrition but this point should not be emphasized. Interviews to understand why people stayed engaged and other did not would be helpful.
  • Elaborating on the point about how to sustain behavior change beyond the program is absolutely critical.
  • More thought should be given to unintended consequences – availability of more money to “eat out” – how does that affect overall health and healthy eating choices?
  • In section 4.5 the authors speak about changed to the enrollment practice to target more participants “with the capacity to be more highly engaged in future iterations.” It is important to understand exactly how this was done and clarify its affect in biasing your sample. Sampling with a bias like this can worsen disparities in larger trials.
  • Not sure the paragraph about focusing on young women is relevant given the objective of this intervention.
  • Conclusion: cannot say it increased fruit and vegetable intake if results not statistically significant.

Reviewer 2 Report

This mixed-methods study assessed the feasibility and explore the effectiveness of the FlipRx program, a prescription produce home delivery and nutrition education program. 

The title and the abstract do not reflect this purpose.  The title should include that this was  a feasibility study and some measure of feasibility should be indicated in the abstract.

Methods

The details of the FlipRx program components need to be described including week by week what was delivered to each  participant and what items were included in the assessment of program feasibility.  I was not clear if every participant received all text message or videos. Was there monitoring of participation by which components were received/viewed? How was engagement assessed? did they have to respond to each monthly survey?

I was not able to discern from the method section how retention was measured. A retention of 60% at 12 months is that the number of people who answered the monthly survey or received the produce? Who are finishers vs non-finishers?

Attrition was mentioned, did anyone ask to stop?

In the qualitative interviews, there was a lot of discussion about use of foods provided. Was there a measure of food waste? Did participants mention not using food in the box?

How did your assess effectiveness?  Was it behavior change or was it acceptability and use of the program materials? Was it a combination of qualitative and quantitative results.

COVID-19 and FI measure.  Any concern about access to food being impacted directly by fear of going to the store?

This mixed methods  only had 25 participants without a comparison group for the quantitative pre- post portion.  This part of the study was not powered to detect differences in food security or dietary intake.  Without a comparison group it is hard to determine if the increase in intake was due to the FlipRx program or a time trend.  The qualitative data provides more indications of effectiveness in changing perceptions.

Attrition - 40% was mentioned. Who dropped out and were they different from those who stayed in the study? The authors mention that the attrition group was not different from the group that stayed in the study.  However, as mentioned many times, this study was not powered to detect differences between groups. It would have been good to include the characteristics of the two groups. It might have also helped to identify a group to target incentives toward in the future.

The quantitative study design was a limitation.  It was a pre-post design without a comparison group.  It is not designed to test causality and it is difficult to determine if the changes observed were attributable to the FlipRx program or some other temporal trend.

The first lines (Lines 602-603) in the conclusion “We found the FLiPRx 12-month home-delivery produce prescription and virtual nutrition education program improved perceived household food security and increased adult and child fruit and vegetable intake.”  This is not true.  The household food insecurity score changes and the dietary intake changes although they suggested improvement, they were not statistically significant and there was not a comparison group and causality cannot be determined.

This study results and discussion section focus on the quantitative results and less so on the feasibility and may have overstated the importance of the quantitative findings and spent more time on the information synthesized between quantitative and qualitative and the FlipRx program feasibility and effectiveness.

Round 2

Reviewer 1 Report

Thank you for the opportunity to review the revised manuscript. The manuscript has been significantly improved and I appreciate all of the edits and efforts made by the authors. I have a few remaining minor edits/issues that I will outline below. One overarching issue I’ve had is the use of the terminology of “perceived” food insecurity. I’m not familiar with this terminology and would instead urge the team to consider referring to a sense of food security experienced by families.  

Title:

Consider changing the title to:

Feasibility of a Home-Delivery Produce Prescription Program to address Food Insecurity

Abstract:

Line 21: would not use “our”

Line 30-34:
this idea of "potential for improved food insecurity" is confusing. Are you saying that qualitative data theme of "improved FI." i would reframe that as families felt more food secure even though there was no change in the FI score?
would suggest using a more scientific term than "well-liked"

line 33 remove However. Reword: further study, with larger sample size, is needed to understand factors influencing participation and assessing effectiveness

Introduction

Last paragraph: would state clearly your hypothesis – the intervention will reduce food insecurity and/or increase fruit and vegetable intake. How are the two related? Making it clear hear so that you can return to this in your discussion.

Methods

Line 106 spelling mistake

Was there a rationale for enrolling only 25?

Line 115: you call it an evidence-based nutrition program. Provide some information on what the cultural adaptation entailed.

Results
line 340 was this average videos viewed PER PARTICPANT? Please clarify.

Line 34-342 can you clarify why the proportions for attendance are with different denominators? (25 and 18?)

Table 1: the formatting can be improved. Usually I see one column that has the variable and the description in one then the column that says “response” should read n (%)

Figure 1: would label the x axis as food insecurity

Figure 2: would label the x axis as fruit INTAKE and vegetable INTAKE

Figure 3: label the x-axis cup of…

Line 480. Note that differences were not significant because sample size is very small

Section 3.9:

The theme of perceived improved FI is still difficult to fully understand. As noted above, consider rewording this as reduced anxiety around food access? Not sure if there is something that speaks to this in the relevant quotes that could convey this meaning in a better way. The issue is that food insecurity isn’t what participants think of it as – is it perceived or are we just not capturing it numerically?

Line 502-503 belongs in the discussion

Discussion

Line 521: alternate word to “well-liked”

Line 528-530: is your exploratory hypothesis just Fruit and vegetable intake? What about food insecurity? What is the pathway you are exploring? Tying this back to the introduction may be important.  You speak about “select group” if this was part of the hypothesis making that clear in the introduction would be helpful.

Reviewer 2 Report

I feel that the authors have addressed my comments in the previous review .

Author Response

Thank you. (No response needed)